# Research on the Effect of Tip Surface Coatings on High-Speed Spindles' Noise

**Hechun Yu [1], Wenchao Li [1], Jin Wang [1,\*], Suxiang Zhang [1], Xiucheng Cao [2], Renzong Wang [3], Guoqing Zhang [1] and Xiaolong Yin [1]**

1   School of Mechatronics Engineering, Zhongyuan University of Technology, Zhengzhou 450007, China; yuhechun@zut.edu.cn (H.Y.); 2020104077@zut.edu.cn (W.L.); 3968@zut.edu.cn (S.Z.); zgq@zut.edu.cn (G.Z.); 6867@zut.edu.cn (X.Y.)
2   Henan NO. 2 Textile Machinery Co., Ltd., Xinyang 464000, China; cxc2609@126.com
3   College of Mechanical Engineering, Donghua University, Shanghai 201620, China; wangrz@zut.edu.cn
\*   Correspondence: 6722@zut.edu.cn

**Abstract:** The contact interface between the stator and the rotor tip of the spindle could be destructed when the spindle is rotating continually at high speed, which will cause strong noise and severe vibration. In order to reduce the sound pressure level of the noise generated by the rotating spindle, three different coating materials, that is, Al-Ti-Cr-C, Ti-C and DLC, were applied to the rotor tip surface of the spindle. The effects of the coating materials on the sound pressure levels of the rotating spindles were studied by using the treated spindles and the untreated spindles. Results showed that compared with Coating Al-Ti-Cr-C, the Coating Ti-C containing only the two main elements of Ti and C produces the smallest sound pressure level in the experiment speed range; the surface roughness of Coating DLC is smaller, but the sound pressure level of the entire spindle becomes larger than Coating Ti-C; the sound pressure level of the spindles with surface coating treatment is obviously lower than that of the spindles without coating treatment. The research results can provide basic data for the design and production of noiseless spinning spindles.

**Keywords:** noise; spindle; AlTiCrC coatings; TiC coatings; DLC coatings; sound pressure level



## 1. Introduction

Noise exposure in the textile industry is a major occupational hazard factor. According to statistics, the range of environmental sound levels from a cotton spinning unit (containing 10,000–20,000 spindles) was from 86 to 95 dB(A), which could make the long-term employees suffer various complications, such as headaches, impaired visual acuity, reduced concentration and reduced reaction speed [1]. The rotation of the spindles is the key factor in inducing environmental noise. Thus, it is important to study the environmental noise induced by the spindles for noise control and employee health.

The textile spindle noise is mainly affected by the design and manufacture defects of the spindle itself, the relative friction between the stator and the tip of the spindle during operation, and the disturbance of the surrounding air induced by vibration when the spindle rotates at a high speed [2]. First, in the design and manufacture stage of the spindle, relevant scholars had studied the spindle and its parts with similar structures and found that the vibration and noise generated during operation can be reduced by changing the geometric parameters [3–6], materials and manufacturing accuracy of the workpiece parts [7]. Additionally, during the operation of the spindle, by improving the lubrication conditions, such as the friction coefficient of the spindle [8–10], and reducing the rotational speed [11,12], the noise generated can be reduced to a certain extent. For example, Wu et al. [8] conducted related experiments in view of the influencing factors of the friction noise induced by a friction pair during rotating and sliding processes and found that the friction noise decreased when the surface roughness decreased; when the relative motion

speed increased, the friction noise increased gradually, but as the rotating speed increased, the friction noise decreased. Nam, J. et al. [10] studied the relationship between the friction coefficient and friction noise, and it was found that when the friction coefficient was less than 0.15 and there was no debris from obvious wear in the contact area, no friction noise was generated. Li et al. [13] studied the influence of polytetrafluoroethylene (PTFE) on the wear and vibration behaviors of modified thermoplastic polyurethane (TPU) sliding against the ZCuSn10Zn2 ring-plates for the vibration and noise of water-lubricated stern tube polymer bearings. The results showed that the lower coefficients of friction (COFs) of modified TPU displayed a small fluctuation amplitude and eliminated vibration waveforms at high vibrational frequencies, which was useful for reducing frictional vibration and noise. Moreover, the factors such as working load [14], friction couple [15] during the running processes will also affect the workpiece noise characteristics. Feng et al. [16] explored the effect of adding Y, Ce, and Ta elements to GLC on the operating noise of gear pumps at different speeds and found that compared with uncoated gears, the noise for the coated gears was decreased by 10 dB at 2000 rpm; good high-frequency noise reduction was obtained, which could be decreased by 19 dB in the range of 4000–10,000 Hz. However, little research focused on reducing the spinning spindle noise by using the method of surface modification.

In recent years, the technology of plasma spraying on the surface of the workpiece is increasingly mature, many scientific researchers who study surface coating materials, preparation methods, coating performance and so on have conducted a lot of research [17,18], and the results are widely used in medical, aerospace, mechanical manufacture and other fields. In the spinning spindle design and manufacturing field, adding surface coating is an effective method to reduce the operating noise of spindles. This is because by adding some specific coatings on the surface of the spindle rotor tip, the strong vibration and severe wear of the spindle can be avoided so as to reduce the spindle noise during the spinning process. However, the effects of the surface coating technology on the spinning spindle noise are manifold.

To reduce the running noise of spinning spindles and improve the working environment of textile workers, three coatings of Al-Ti-Cr-C, Ti-C and DLC are deposited on the surface of the ingot tip, and the noise performance of spindles treated with Al-Ti-Cr-C, Ti-C and DLC coatings were compared with the untreated spindles. The sound pressure levels of various treated and untreated spindles during the normal operation were studied. The research results can provide a research basis for the design and manufacture of spinning spindles with low sound levels and the development of the spinning workshop towards a noiseless direction.

## 2. Experiment Methods and Equipment

### 2.1. Samples and Parameters

Based on the high-frequency (HF) type structure principle, the YD4203 spindles that were provided by Henan NO. 2 Textile Machinery Co., Ltd. (Xinyang, China), were used in these experiments, and the longitudinal buffer element was prepared. It has good vertical cushioning and a horizontal auto-centering function. The advantages of these spindles are their strong load-bearing capacity, small amplitude and long life. However, this type of spindle will generate a lot of noise when rotating at a high speed, and the average noise generated by a single spindle running at 20,000 r/min is as high as 74 dB [19], which will seriously affect the physical and mental health of the textile workers, so the study of the noise reduction method of the spindles is necessary.

Since the Al-Ti-Cr-C [20], Ti-C [21] and DLC [22] coatings showed good wear-resistant and lubricating properties, these coatings were chosen and constructed on the surface of the spindles using the surface cladding technology in this study.

Although the recommended working speed range of YD4203 spindles is 16,000 r/min~20,000 r/min, the spindle speed would gradually increase from a standstill to a value larger than the recommended speed due to inertia action. Thus, the rotating

speed of the spindles in this experiment ranged from 1000 r/min to 21,000 r/min, with 1000 r/min as a tolerance. At all of the above speeds, the average amplitude and sound pressure levels of the noise generated by the spindles with three surface coatings (Al-Ti-Cr-C, Ti-C and DLC coatings) were collected. Additionally, the time and SPL (Sound Pressure Level) curves had been drawn to study the influence of the surface coating of the spindle on its noise during work comprehensively.

### 2.2. Coatings on the Rotor Tip Surfaces of Spindles

The coatings were electro-spark deposited onto the spinning spindle substrate by means of the Spark Depo MODEL 300 made by TechnoCoat Co.,Ltd.( Nagoya, Japan). The deposition parameters were chosen based on the analyses of the current characteristics as well as the manufacturer's recommendations. Before the deposition, the vacuum chamber was pumped out to a pressure of $3 \times 10^{-3}$ Pa. The ESD layer deposition was carried out in an Ar atmosphere at a constant pressure of 1 Pa. Then, the depositions of the electro-spark deposition layers were performed at an electrode pulsed voltage of 25 V, a pulse frequency of 650 Hz, a width of 80 μs, and a pulse current of 120 A, respectively. Electrode movement parameters were set as follows: the rotation speed is 1000 r/min, the scan rate is 500 cm/min, and the scanning step is 0.5 mm.

The unit cells of these coatings are shown in Figure 1. The mixed powders were pre-placed on high-speed spindles. The structure and surface morphology of the spindles with three different coatings are shown in Figures 2 and 3. The complete 2D technical drawing of the experimental spindle is shown in Figure S1 in Supplementary Materials.

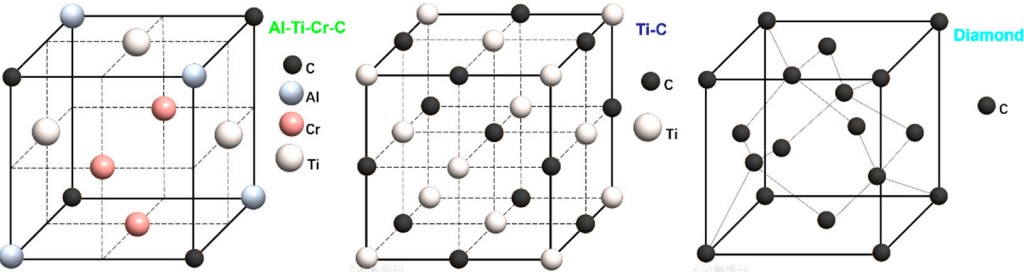

**Figure 1.** Crystal Structure Models of AlTiCrC, TiC, and Diamond.

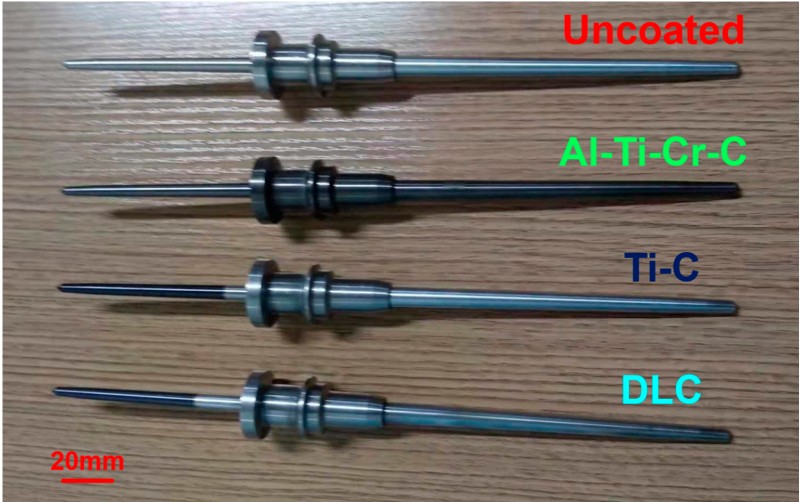

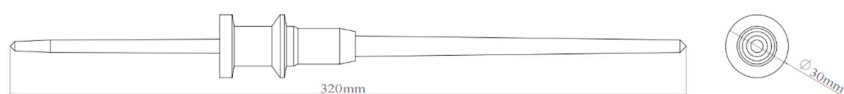

**Figure 2.** Experimental spindle and its 2D technical drawing.

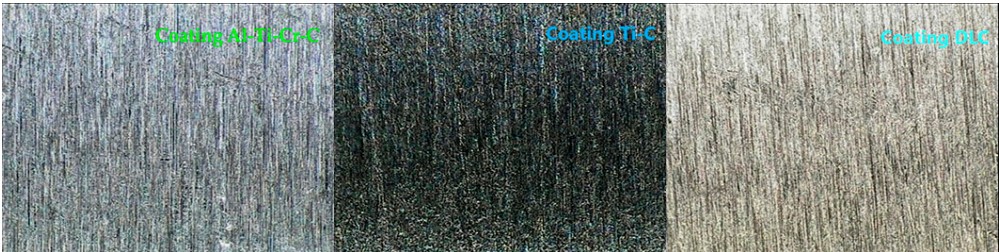

**Figure 3.** Surface morphology of the spindles.

## 2.3. Coatings Characteristics Tests

The single side thickness and surface roughness were tested using a 3D Measuring Laser Microscope (LEXT OLS4000, OLYMPUS, Miyazaki, Japan). The 3D Measuring Laser Microscope is shown in Figure 4. For the coating thickness test, the tips of three spindles with different coatings were embedded in plasticine and put on the testing bench. The microscopic structures of the tips of three spindles with different coatings were observed and analyzed at 100× magnification. For the surface roughness test, three intact spinning spindles with different coatings were fixed on a special fixture and observed at 100× magnification. The whole process is repeated five times. Finally, the software of the equipment was used to obtain the average thickness and roughness data of the coating through the collected microscopic images.

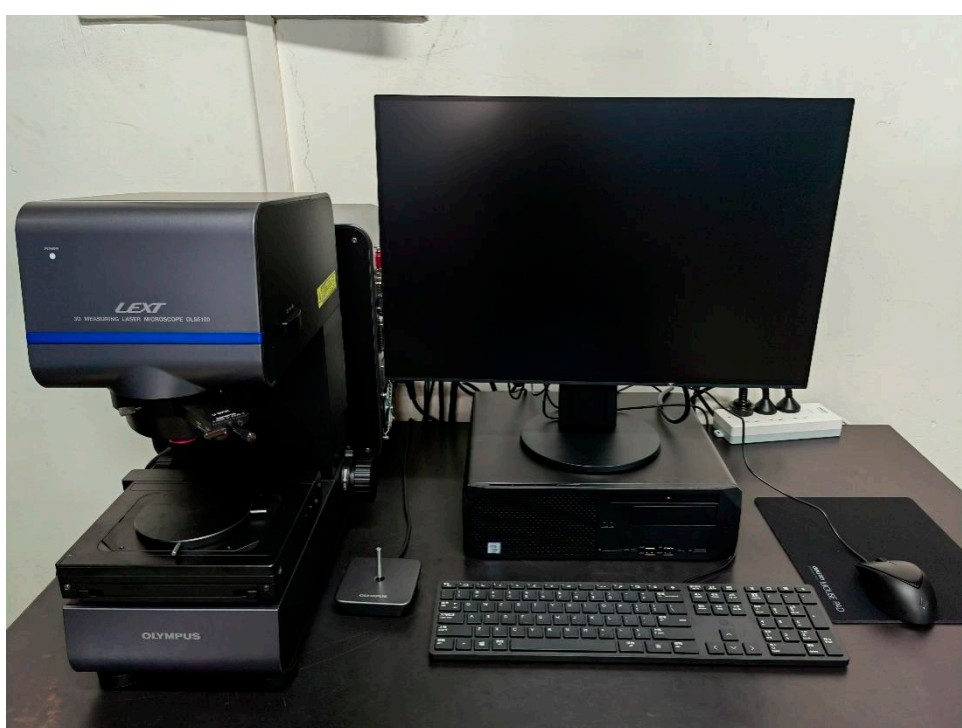

**Figure 4.** Three-dimensional Measuring Laser Microscope.

The hardness and coefficient of friction were tested using the Automatic Hardness Tester (VH3300 Wilson, Lake Bluff, IL, USA) and Rolling Wear Tester (HS-5001, Dedu instruments, Henan, China), respectively. The hardness test was conducted using the tips of three spindles with different coatings under the force of 100 N at 40× magnification. Ten test points are selected for the measurement. The coefficients of friction between the tips of the spindles and bottom benches were tested under a loading force of 5 N and a rotating speed of 2000 r/min. The rotating duration was 3 min.

All of the above tests were conducted in a clean laboratory with a room temperature of 25 °C ± 2 °C and humidity of 55% ± 4%.

### 2.4. Experiment Setup and Data Processing

In order to accurately measure the rotation speed of the spindle in the experiment and the sound pressure level of the noise generated at different rotation speeds, the multi-functional spindle detection system [23] with a speed up to 60,000 r/min. In this study, a 0~2 mm vibration measuring instrument was used with an accuracy of 0.1 μm. A sound pressure level measurement of 25~130 dB and an accuracy of 0.1 dB were used in this study, as shown in Figure 5. The spindles used in the experiment were measured at different speeds, and the SPL at different speeds was obtained. The complete 2D technical drawing of the multi-functional spindle detection system can be found in Figure S2 in Supplementary Materials.

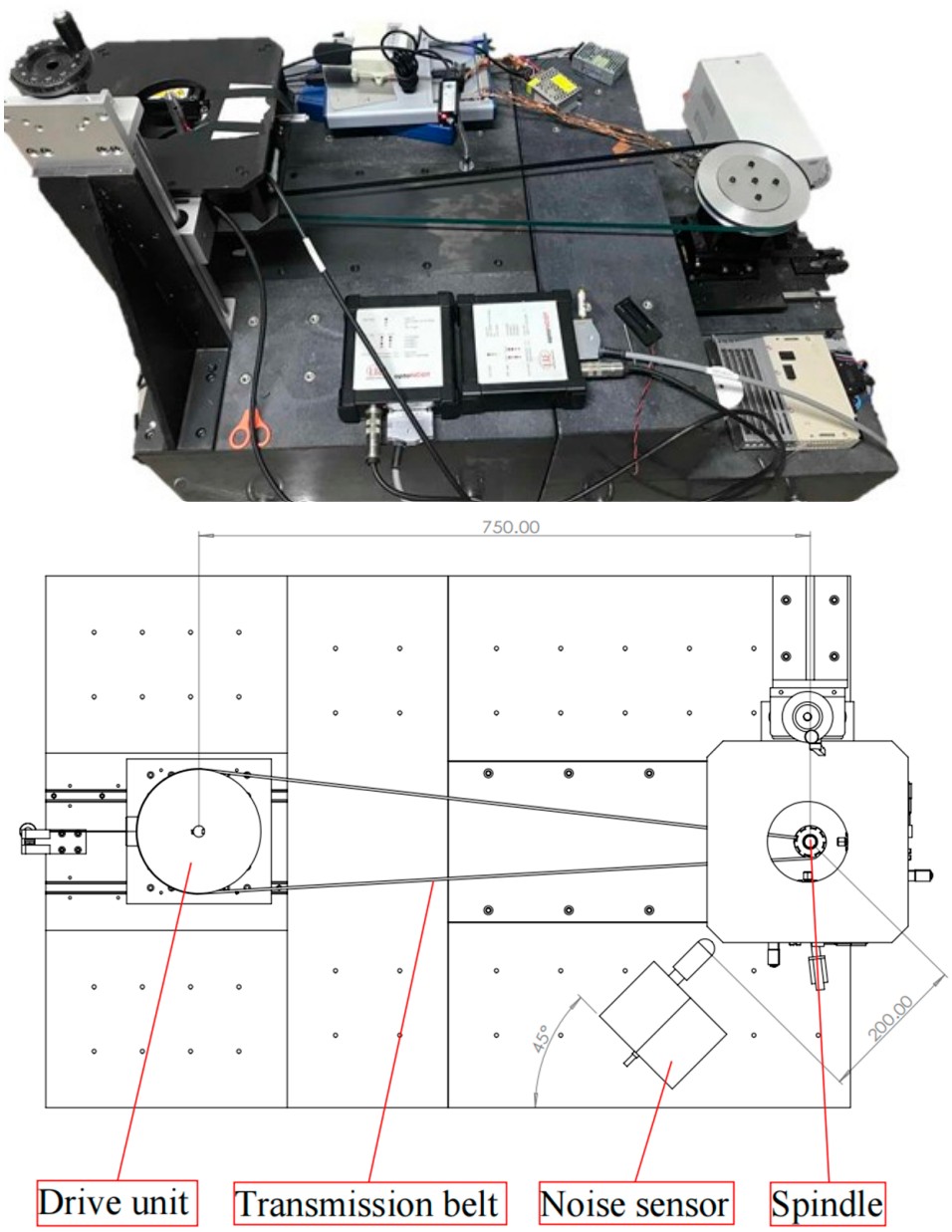

**Figure 5.** *Cont.*

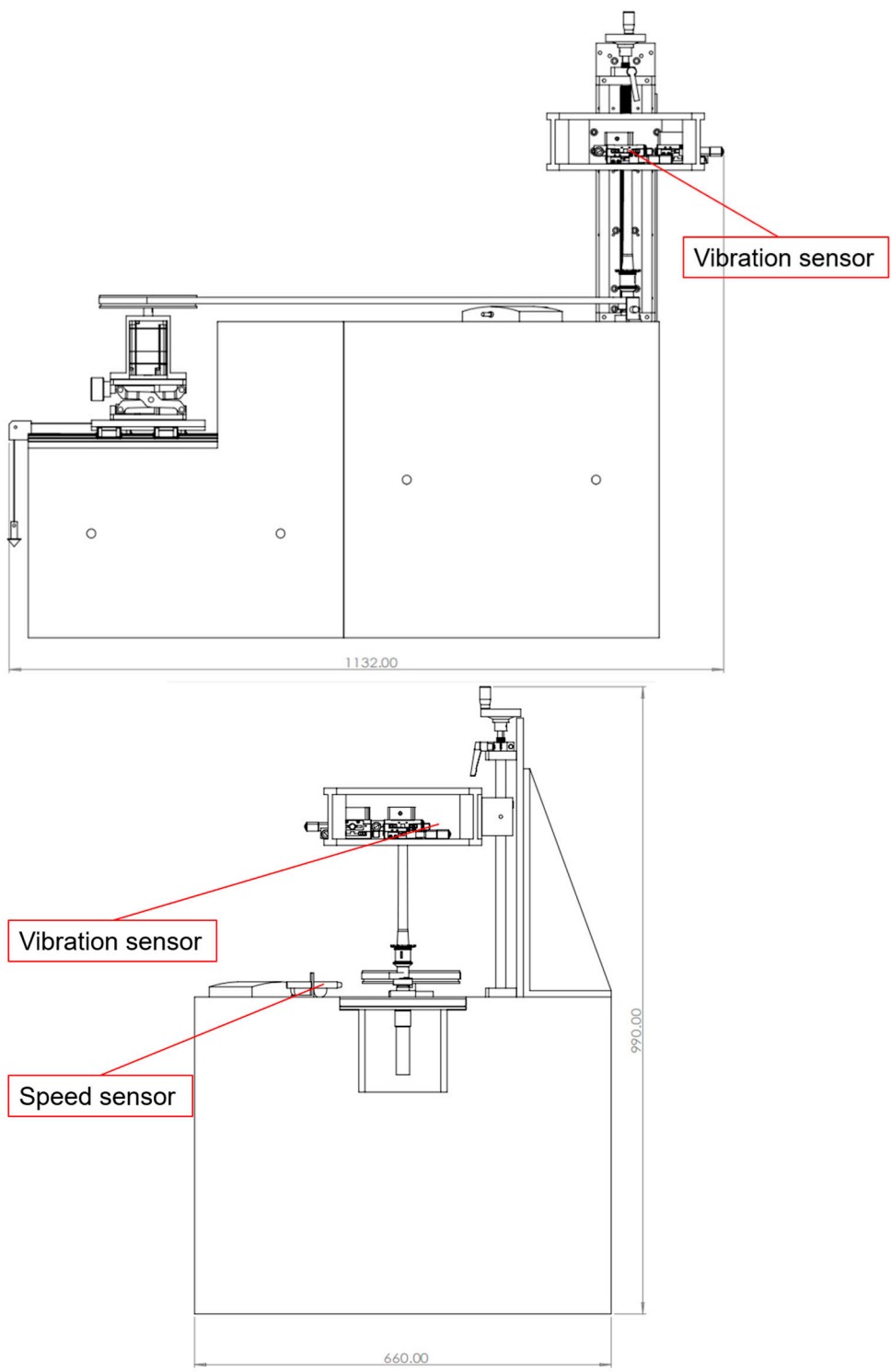

**Figure 5.** Multifunctional spindle inspection system.

The experimental instrument data used in this study in the multi-functional spindle detection system are shown in Table 1.

**Table 1.** The multi-functional spindle detection system.

| Instrument Type | Model | Manufacturer Name | Country for Instruments | Range | Precision |
|---|---|---|---|---|---|
| motor | SGM7A-04A | YASKAWA | Kitakyushu, Japan | - | - |
| noise sensor | RS-ZS-V10-2 | Shandong Renke Control Technology | Shandong, China | 25~130 dB | 0.1 dB |
| Speed sensor | ROS-P | Monarch Instrument | Amherst, NH, USA | 1–250,000 r/min | ±0.0015% |
| Vibration sensor | OptoNCDT 2200-2 | Micro-Epsilon | Ortenburg, Bayern, Germany | 0–2 mm | 0.1 μm |
| Multifunction I/O Device | USB-6361 | National Instruments | Austin, TX, USA | - | - |

During the noise experiment, the speed signal can be directly collected using a counter and according to the algorithm inside the experiment system, but the sound pressure level signal is converted into an electrical signal during the collection process, so it is necessary to rely on a formula to convert the electrical signal into the sound again. The conversion formula of voltage level signal is as follows:

$$SPL = 20 * log_{10}\left(\frac{V}{V_0}\right) \tag{1}$$

in which $V$ is the experimentally measured electrical signal value; $V_0$ is commonly used sound pressure corresponding to electrical signal value.

*2.5. Vibration and Noise Experiment Procedure*

The lubricating oil used in the experiment is DERUKO No. 10 spindle lubricating oil, the density at 288 K is 0.827, the kinematic viscosity (313 K°C mm$^2$/s) is 10.5, the flash point is 438 K, and the pour point is 255 K.

Before the noise experiment, the above four kinds of spindles were subjected to vibration experiments at different rotational speeds. In the vibration test, the sampling rate is set to 5000 Hz, and the sampling number is 10,000. During sampling, each spindle was sampled five times at each rotational speed to obtain three sets of experimental data for the same spindle at the same rotational speed. That is, the number of repeated tests for each coated spindle is 15 times. The results of the vibration test can be used to analyze the vibration noise generated by the spindle during operation.

The noise experiment sound pressure level data was collected after the rotation speed was close to the predetermined rotation speed, and the change was small within a period of time. In order to reduce the influence of external environment factors on the collected experiment data, the sampling rate was set to 10,000 Hz, and the number of samples was 10,000; that is, the time interval between two adjacent data collections is only $10^{-4}$ s. When sampling, each spindle was sampled five times at each rotating speed, and three sets of experimental data of the same spindle at the same speed were obtained. That is, the number of repeated tests for each coated spindle is 15. After confirming that there was no obvious external environment factor, the average value and standard deviation of all the converted data, as well as the maximum and the minimum values, were calculated for analysis.

Because the sound pressure level of the spindles fluctuates within a certain range, the fluctuation range was relatively large, as shown in Figure 6a. To display the sound pressure level during the rotation of each spindle more clearly in the time-domain graph, the original sound pressure level image was fitted by smoothing filtering using Origin software (OriginLab, Northampton, MA, USA), and the variation trend of the spindle sound pressure level was obtained, as shown in Figure 6b.

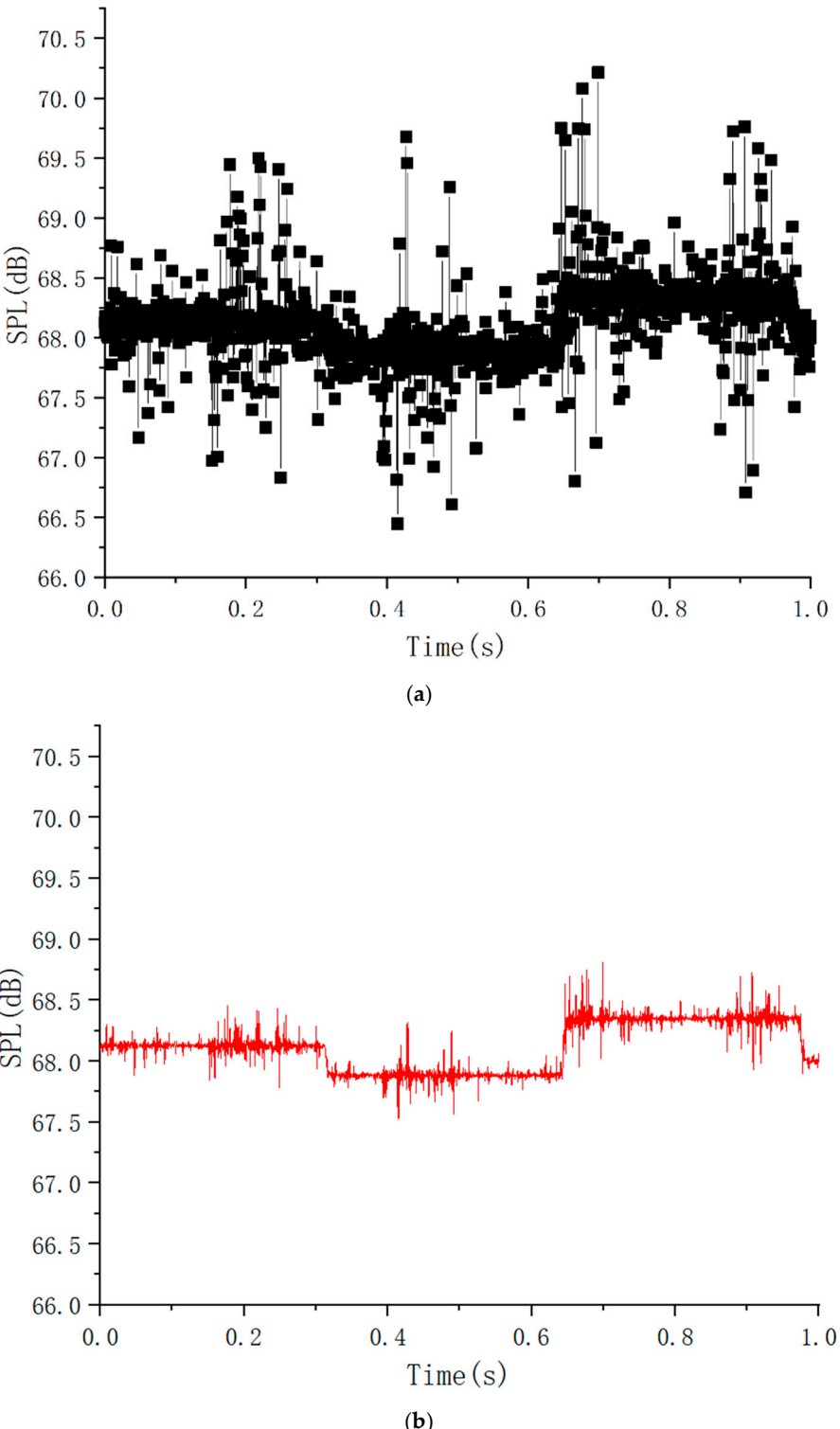

**Figure 6.** FFT fitting of sound pressure level: (**a**) original time domain diagram of sound pressure level; (**b**) sound pressure level after smooth application.

The spindle without surface coating structure was used as the control group for comparison purposes, and the sound pressure levels of these surface-coated spindles compared with the original spindle at each experimental speed were analyzed to explore the noise performance of these surface coatings.

## 3. Results and Discussion

### 3.1. Coating Inspection and Lubrication Calculation

The measured mechanical and tribological properties of the spindle are shown in Table 2. It can be seen from Table 2 that the Ra values of the three coated spindles have little difference; Coating Al-Ti-Cr-C has the highest hardness, and Coating DLC has the lowest hardness. Coating Al-Ti-Cr-C has the largest coefficient of friction. The Coating DLC has the smallest coefficient of friction.

**Table 2.** Types of coatings and their mechanical and tribological properties.

| Surface Coating Type | Main Chemical Composition and Proportions | Single Side Thickness (μm) | Surface Roughness (Ra) (μm) | Hardness (HV) | Coefficient of Friction (−) |
|---|---|---|---|---|---|
| Coating Al-Ti-Cr-C | Al(20%), Ti(50%), Cr(20%), C(10%) | $3 \pm 0.1$ | $0.376 \pm 0.002$ | $3500 \pm 20$ | 0.3~0.35 |
| Coating Ti-C | Ti(80%), C(20%) | $3 \pm 0.1$ | $0.377 \pm 0.002$ | $2800 \pm 18$ | 0.1~0.2 |
| Coating DLC | sp2(80%), sp3(20%) | $2.5 \pm 0.1$ | $0.377 \pm 0.002$ | $2600 \pm 16$ | 0.06~0.1 |

The calculation of lubrication properties for rough surfaces using general Reynolds:

$$\frac{d}{dx}\left(\frac{h^3}{12\eta}\frac{\partial \overline{p}}{\partial x}\right) + \frac{d}{dy}\left(\frac{h^3}{12\eta}\frac{\partial \overline{p}}{\partial y}\right) = \frac{\partial}{\partial x}\left(\frac{h(U_2 - U_1)}{2}\right) + \frac{\partial}{\partial x}\left(\frac{h(V_2 - V_1)}{2}\right) + \frac{\partial h}{\partial x} \quad (2)$$

in which $p$ is the fluid film pressure; $h$ is film thickness; $x$ and $y$ are the bearing length and width; $\eta$ is the viscosity of the fluid; $U_2$ and $U_1$ is the velocity in the x direction; $V_2$ and $V_1$ is the velocity in the y direction.

The formula for calculating the oil film thickness $h$ is:

$$h = h_0 + h_s \quad (3)$$

In which $h_0$ is the distance between the centerlines of the two surface profiles, which is the deterministic part of the oil film thickness; $h_s$ is the deviation between the actual film thickness and the nominal oil film thickness due to the influence of surface roughness.

Using Matlab to calculate the Reynolds equation, the results are shown in Figure 7. See Solution_techniques_for_Reynolds_equation in Supplementary Materials for MatLab files.

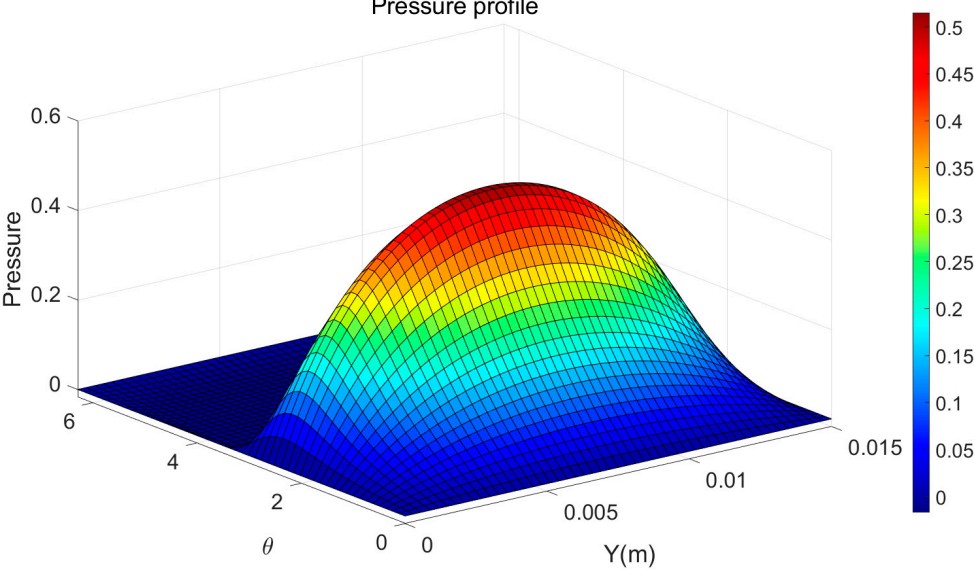

**Figure 7.** Reynolds equation calculation results.

Since the difference in surface roughness of the three types of coated spindles is small and other variables are the same, the calculated oil film thickness difference is small. In addition, because the difference in oil film thickness of the three types of coated spindles is very small, it is difficult to obtain results that have a significant impact on spindle lubrication from the Reynolds equation solution.

### 3.2. Vibration Pre-Experiment Results and Analysis

Table 3 shows the average amplitudes of the four spindles at different experimental speeds. At the rotational speed of the vibration experiment, the average amplitudes of the three types of coated spindles are all smaller than those of the uncoated spindles; the average amplitude of the coated Al-Ti-Cr-C varies greatly, the maximum amplitude is twice the minimum amplitude, while The amplitude changes of other coated spindles are small; when the uncoated spindles are not considered, at most rotating speeds, the amplitude generated by the Coating Ti-C is the smallest, and the amplitude generated by the Coating Al-Ti-Cr-C is the largest. The vibration damping effect of layer Ti-C is the best. The results of the high-speed spindle vibration experiment can be used to explain some phenomena in the noise experiment.

**Table 3.** The change graph of rotation amplitude-speed.

| Speed (r/min) | Uncoated Average Amplitude (mm) | Al-Ti-Cr-C Average Amplitude (mm) | Ti-C Average Amplitude (mm) | DLC Average Amplitude (mm) |
|---|---|---|---|---|
| 1000 | 0.009 | 0.008 | 0.007 | 0.007 |
| 3000 | 0.011 | 0.012 | 0.008 | 0.009 |
| 5000 | 0.009 | 0.008 | 0.007 | 0.009 |
| 7000 | 0.008 | 0.007 | 0.005 | 0.006 |
| 9000 | 0.011 | 0.007 | 0.008 | 0.005 |
| 11,000 | 0.012 | 0.012 | 0.009 | 0.006 |
| 13,000 | 0.010 | 0.011 | 0.009 | 0.007 |
| 15,000 | 0.008 | 0.010 | 0.008 | 0.008 |
| 17,000 | 0.009 | 0.009 | 0.008 | 0.009 |
| 19,000 | 0.007 | 0.006 | 0.006 | 0.008 |

### 3.3. The Sound Pressure Levels of Rotating Spindles

The changes in the sound pressure level of the four kinds of spindles with the speed and time are shown in Figure 8. The sound pressure level of the four kinds of spindles basically increases with the increase in speed and fluctuates within a certain range. When the spindle was subjected to a large amount of external disturbance during operation, the sound pressure level would have a wide range of changes, which usually lasts for a very short time and may be beneficial or unhelpful in noise reduction.

When the rotation speed is large, that is, the rotation speed is in the range of 12,000 r/min~21,000 r/min, the stable hydrodynamic lubrication is formed between the friction surfaces of the spindle so that when the spindle speed increases, the change in the sound pressure level is small. This is similar to the results of Wu et al. [8]. It can be seen that when the rotational speed of the spindle is less than 7000 r/min, the rotational speed has a great influence on the SPL generated during operation.

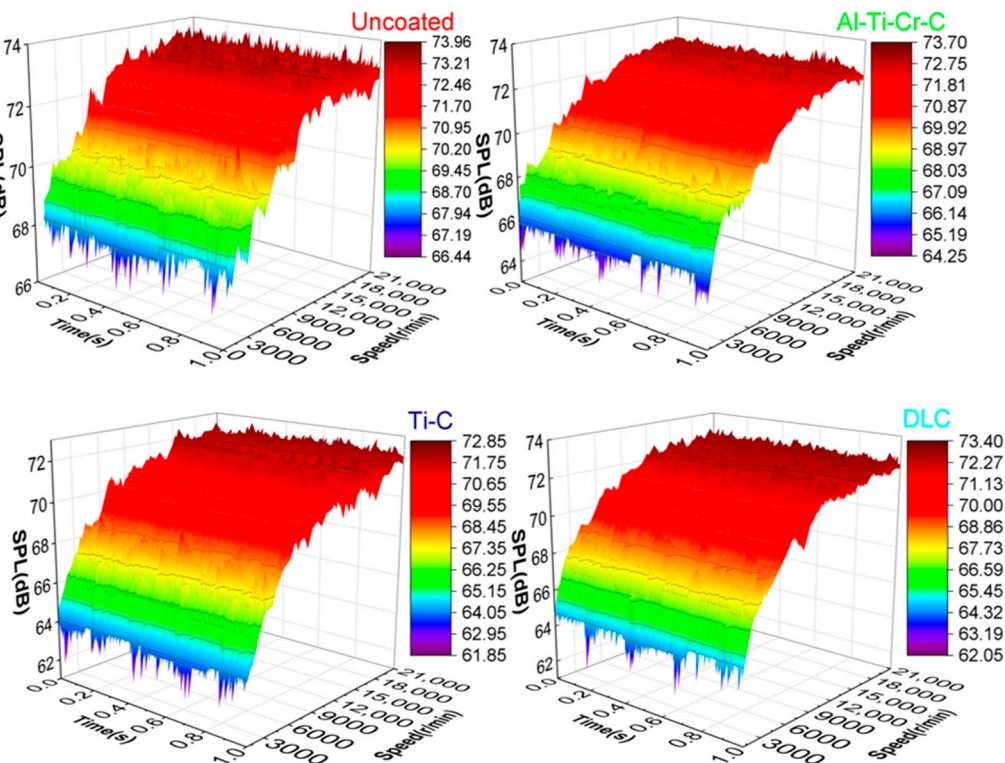

**Figure 8.** The change graph of rotation speed–time–sound pressure level.

*3.4. Comparison of Spindle Sound Pressure Level at the Same Speed*

It can be seen from Figure 9 that no matter how high the rotating speed of the spindle is, at the same rotating speed, the sound pressure level of the Coating Ti-C spindle during rotating is the smallest, while the sound pressure level of the untreated spindle is basically higher than that of the other three spindles. When the rotating speed of the spindle is no more than 18,000 r/min, at the same rotating speed, the differences between the sound pressure level of the rotating of the untreated spindle and the three kinds of coated spindles are significant. In the range of 17,000–20,000 r/min, with the increase in the speed, the sound pressure level difference of several coated spindles at the same speed decreases gradually; when the speed is higher than the recommended working speed range and reaches 21,000 r/min, the difference in sound pressure level is significantly larger than that of 19,000 r/min and 20,000 r/min.

Compared with the Coating Al-Ti-Cr-C, the Ti-C contents in the Coating Ti-C are relatively high, and Al and the Cr elements form $Al_2O_3$ and $Cr_2O_3$ mixed oxide film on the coating surface. As a result, the friction coefficient of the Coating Al-Ti-Cr-C spindle is large, leading to a larger sound pressure level of the spindle during operation. This result is similar to that of Li Yang et al. [24], who studied the effect of Al/Cr composite coating on the hot corrosion properties of Ti2AlNb alloy, and Nam, J. et al. [10]'s study of friction coefficient on friction noise.

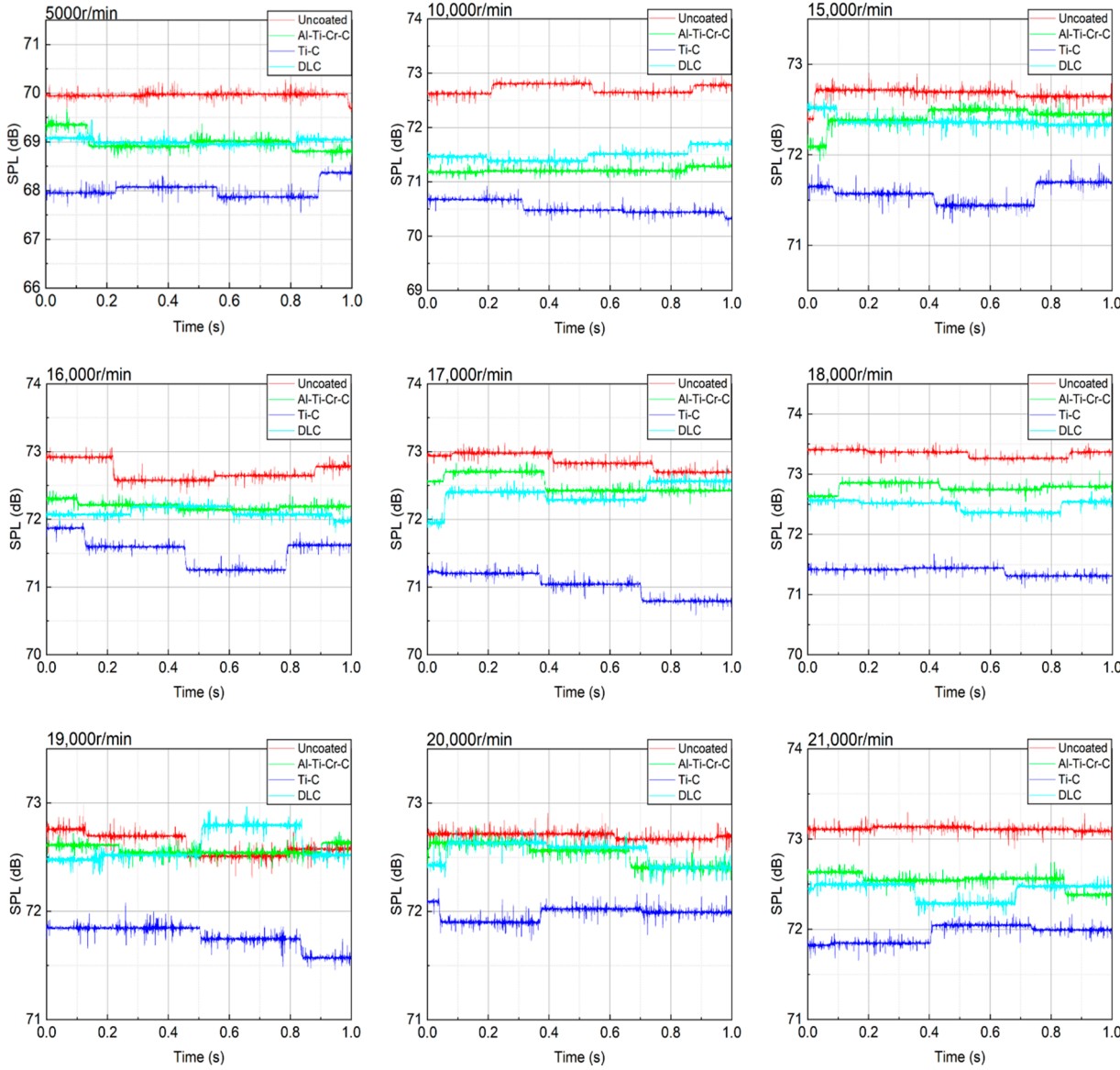

**Figure 9.** Sound pressure levels vs. Times of spindles at each speed.

## 3.5. Overall Comparison of Sound Pressure Levels of Each Spindle

Figure 10 shows the curves of the average sound pressure level of the four spindles. It can be seen that the sound pressure level of the Coating Ti-C spindle at each rotating speed is smaller than that of the other three spindles. The sound pressure level of the Coating Al-Ti-Cr-C spindle is similar to that of the Coating DLC spindle. The sound pressure level of the spindle with the coating is smaller than that of the untreated spindle except when the spindle speeds are 19,000 r/min and 20,000 r/min.

As shown in Figure 11, the comparison of the maximum value of the sound pressure level of the noise produced by the four kinds of spindles at the experiment speed is similar to the average value. Among them, the sound pressure level of the noise generated by Coating Ti-C at multiple experiment speeds is smaller than that generated by other spindles at the same speed.

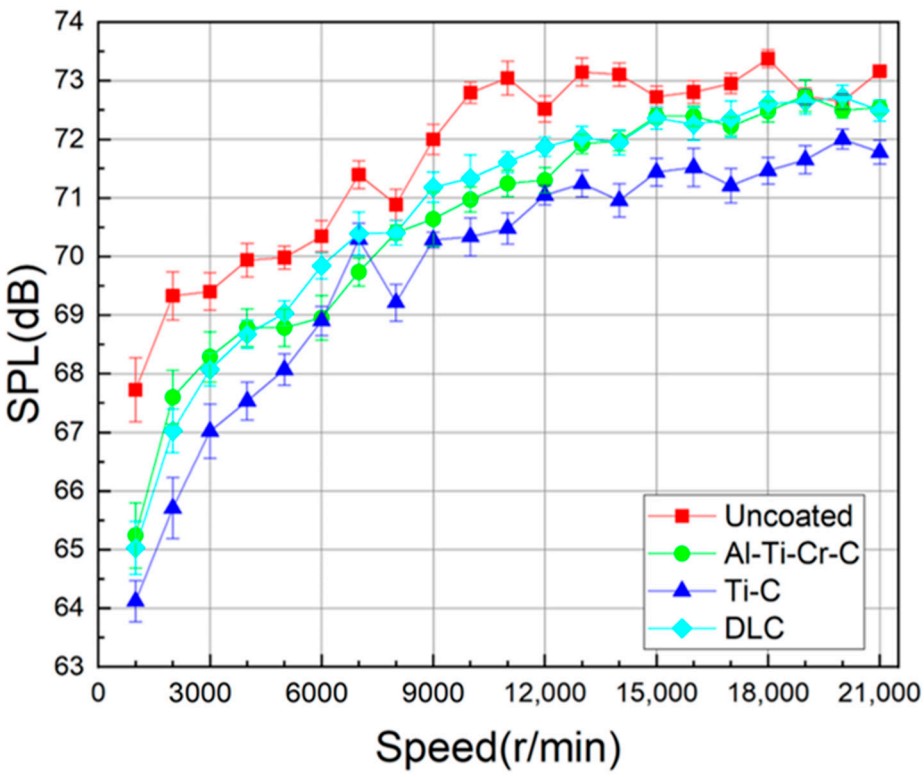

**Figure 10.** Average sound pressure level of the spindle–speed graph.

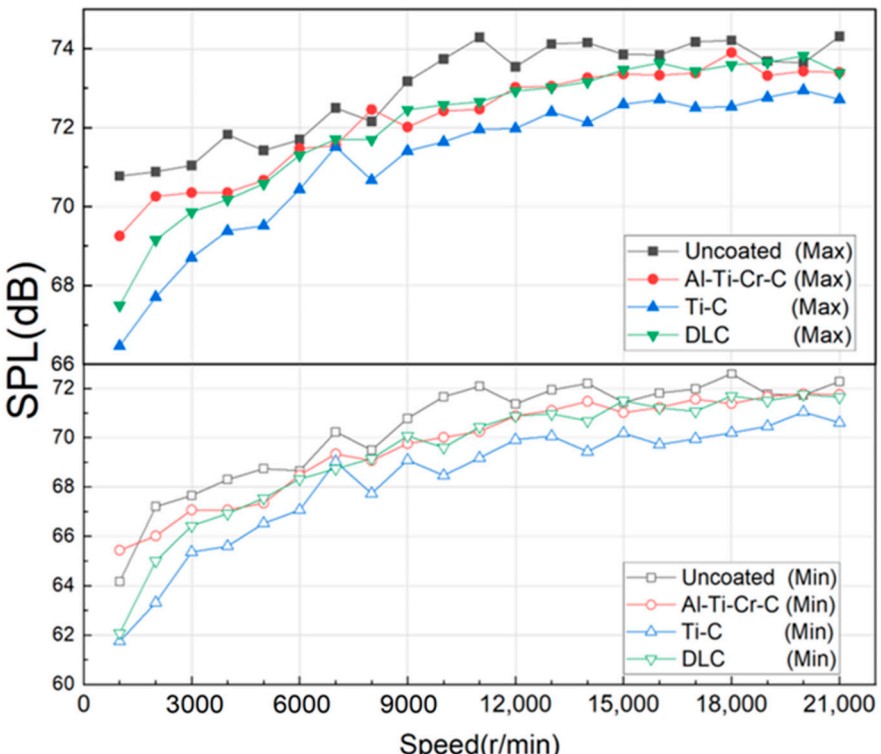

**Figure 11.** Maximum value of the spindle sound pressure level–speed graph.

When the spindle rotates at low speed, due to the formation of metal carbide or diamond-like carbon film on the surface of the coatings, the friction coefficient of the three coated spindles is lower than that of the untreated spindle, and the lubrication conditions

are better. Therefore, the sound pressure level of the coated spindle is higher than the Untreated spindles are large.

The Coating Ti-C elements produce a smaller sound pressure level in the experiment speed range than the Coating Al-Ti-Cr-C and the Coating DLC. The running noise of a high-speed spindle is roughly composed of friction noise and vibration noise. Since the three types of coated spindles and the lubricating oil used are the same, the surface roughness is similar, and according to the calculation results of the Reynolds equation, it can be considered that the lubrication conditions are the same. The spindle vibration experiment (Table 3) shows that the vibration generated by Coating Ti-C during operation is smaller than that of the other two types of coated spindles. It can be seen that Coating Ti-C generates less vibration noise during operation, which makes the overall SPL smaller. This is similar to the results of Li et al. [12].

Partially processed experimental data are shown in Table S1 in Supplementary Materials.

At present, the production lines of textile factories are mainly divided according to the blowing and carding equipment. Generally, a factory with 30,000 spindles is divided into two production lines. One is a long-staple cotton production line, which mainly spins high-count yarns of more than 50 counts, with a scale of 20,000 spindles. Another fine-staple cotton production line spins low-count yarns below 40 counts, with a scale of 10,000 spindles. Although the coated spindle used in this paper is only 1–2 dB lower than the uncoated spindle, if the noise generated by 10,000 spindles is reduced by 1–2 dB at the same time, it will improve the working environment of textile workers.

In this study, only three different types of coatings were prepared on the surface of the spindle tip, and there are many types of coatings. It is still unknown whether other coatings have obvious effects on the noise reduction of the spindle. Secondly, in the coatings made in this research, the proportion of powder of each element is a fixed value, and whether the proportion of powder is the best percentage for noise reduction of the spindle also needs to be studied. Thus, further research can be carried out in terms of changing the proportion of different powders and preparing different coatings on the surface of the ingot.

## 4. Conclusions

In this paper, by comparing three kinds of different coated spindles and the untreated spindles, the sound pressure levels of the spindles were collected and analyzed with the rotating speed in a range of 1000 r/min~21,000 r/min, and the following conclusions were reached:

1. Because the Al and the Cr elements form $Al_2O_3$ and $Cr_2O_3$ mixed oxide film on the coating surface, compared with the Coating Al-Ti-Cr-C, the Coating Ti-C containing only the two main elements of Ti and C produces the smallest sound pressure level in the experiment speed range;

2. Although the friction coefficient of Coating DLC is smaller than that of the other two types of coated spindles, it has a larger vibration during operation, resulting in larger vibration noise caused by the vibration of the spindle, so its noise reduction effect on the spindle is not ideal;

3. The sound pressure level of each surface-coated spindle is significantly lower than that of the uncoated spindle, and the noise reduction effect of the Coating Ti-C is the most obvious.

The coating is deposited on the surface of the spindle tip of the high-speed spinning spindle to improve the noise generated by the high-speed spindle during operation. The research results can provide basic data for the designer and manufacturer to design and manufacture new noiseless spinning spindles.

**Supplementary Materials:** The following supporting information can be downloaded at: https://www.mdpi.com/article/10.3390/coatings12060783/s1, Figure S1: Experimental spindle; Figure S2: Multifunctional spindle inspection system; Figure S3: Supplementary Material 1.1; MatLab file: Solution_techniques_for_Reynolds_equation.

**Author Contributions:** Conceptualization, H.Y. and J.W.; methodology, J.W.; software, X.Y.; validation, S.Z., G.Z. and R.W.; formal analysis, W.L.; investigation, J.W.; resources, H.Y. and X.C.; data curation, J.W.; writing—original draft preparation, W.L.; writing—review and editing, J.W.; visualization, J.W.; supervision, H.Y.; project administration, J.W.; funding acquisition, H.Y. All authors have read and agreed to the published version of the manuscript.

**Funding:** This work is supported by the National Natural Science Foundation of China (grant no. 51875586), Colleges and universities in Henan province youth backbone teacher training program (2018GGJS105) and the China National Textile and Apparel Council (grant no. 2017104 and 2019067).

**Institutional Review Board Statement:** Not applicable.

**Informed Consent Statement:** Not applicable.

**Data Availability Statement:** Data sharing is not applicable to this article. Supplementary Material 1.1 Average SPL for the spindle with different coating types under various speeds.

**Conflicts of Interest:** The authors declare no conflict of interest.

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
