# Peer review of "Research on the Effect of Tip Surface Coatings on High-Speed Spindles’ Noise"

_coatings, doi:10.3390/coatings12060783_

Round 1

Reviewer 1 Report

The authors have made an overall revision to the draft based on the first comment, and the results are quite satisfying.

Nevertheless, I still insist the authors provide technical drawing 2D so that all views (isometric, upper, side, lower sides) of the experimental subject can be seen clearly.

Author Response

Dear Editors and Reviewers:

Thanks again for your constructive comments and valuable suggestions. We have carefully revised the manuscript according to the reviewer's suggestions. Our responses to the comments are listed below, and the corrections to the manuscript are in red font.

Reviewer 2 Report

The idea of the paper is interesting and innovative. However, there are several flaws that need to be corrected in order to recommend the paper for publishing. 

  • References 12 and 13 do not describe the plasma spray process. There are many papers and books that theoretically describe the plasma spray parameters and process itself like Vencl A., Optimization of the deposition parameters of thick atmospheric plasma spray coatings, Journal of the Balkan Tribological Association, 18, 3, 2012, 405-414; or Pawlowski L., The Science and Engineering of Thermal Spray Coatings, John Wiley & Sons, Chichester, 2008.
  • In line 71 the authors write “However, this type of spindles will produce a lot of noise when they are rotating at high speed, which will seriously affect the…” Can they give some data on noise level with the references?
  • Table 2 and 3 are not mentioned in the text. How did the authors provide data in Table 1? Did they measure it or..? If they measured it, what were the conditions?
  • Maybe “drive unit” is a better solution instead of “gearing” in Figure 4. In addition, it would be much better if the authors showed the contact region of the spindle and the counter-body so the contact conditions could be presented. The general view of the equipment is interesting but do not give any crucial information.
  • Line 148 “… each spindle was sampled five times at each rotating speed, and three sets of experimental data of the same spindle at the same speed were obtained”. So what was the number of repeated tests per each speed and material: 3 or 5?
  • Please do not use the term “surface friction coefficient” since the coefficient of friction is always “on the surface”.
  • Line 210-211 “…lubrication condition of the Coating SP2\SP3\C+ is not as good as that of the Coating Ti-C spindles..” lubrication conditions are in direct correlations with the COF and Ti-C coatings showed higher COF, so this can not be the explanation.
  • It would be interesting to calculate the percentage decrease in sound level at the highest speeds. I guess it is between 1 and 2%. Please discuss if this decrease is significant enough to apply the coatings at all and is enough to justify the price of the applied coating and deposition process.
  • There is no discussion of the results. The authors give nominal composition, hardness, COF, structure etc. of the coatings but did not make the connection with the obtained sound level (only for COF and it was not completely adequate).
  • What was the purpose of giving Figure 9 if we have the deviations in Figure 8?

Line 229-230 “…the surface roughness of the three kinds of coated spindles was reduced…” but the authors did not provide the results of these measurements.

Author Response

(The authors gave the same response as above.)

Reviewer 3 Report

After the first revision, the work was improved, but not enough. We saw that the DLC roughness is almost the same as for the rest of the coatings (see Table 1). Under these circumstances, all comments about differences in roughness values are speculative.

On the contrary, the DLC hardness is quite surprisingly low. Do the authors find similar values in the literature? The low friction coefficient indicates lower noise results, but this should be commented on considering hardness and not roughness, which is similar for all coatings.

Instead of calculating the lubrication parameter, the authors only presented the Reynolds equation for HD lubrication and no results.

Observing the replies to my previous comments I see that the authors did not perform any other experiments, but tried to introduce some irrelevant information for this paper.

Considering that the noise reduction was only 1-2 dB, I do not think this research is worth publishing.

Author Response

(The authors gave the same response as above.)

Reviewer 4 Report

Dear Authors,

Congratulations on your work, which is focused on a very interesting subject. As any other paper in this phase, there are some amendments to do, whose can improve the overall quality of your paper. Thus, I'm providing below some comments and suggestions, trying to collaborate by this way in improving your paper:

  1. The Abstract doesn't clearly state the literature gap found, as well as the main motivation to develop this work. Thus, please clearly state the gap found in the literature in the Abstract, Introduction and Conclusions. The mains goals are also not clear in the Abstract.
  2. The novelty brought by your work is also not properly pointed out. Thus, please state clearly the novelty that your paper represents for the scientific community, stating as well if your contribution is exclusively scientific or if there was some practical motivation behind the development of your work. Any industrial application based on this work should also be pointed out.
  3. Keywords used should be improved, including the coatings used, as well as others that help finding this article by other Authors.
  4. The number of References used is lower than expected. Moreover, no direct speech is used, describing the main goals of other researchers, the methods used, as well as the main conclusions achieved. These results should help you in the Discussion of your results.
  5. Regarding the References, please consider to use the following ones: DOI: 10.3390/s20164536; DOI: 10.1016/j.promfg.2020.10.036.
  6. Sentences such the one starting 2.2 sub-section need to be supported by literature. Please include one or two references.
  7. Caption of Figure 1 is lacking of information. Please add complete captions in figures and tables, helping the Reader to quickly understand what is seeing and how the image is shown.
  8. Please add a bar scale in Figure 2.
  9. Please include information on many samples have been analysed in each case (coatings).
  10. Caption of Table 1 is not correct because we are talking about properties.
  11. Please include a table with the main coating deposition parameters used, as well as the deposition machine (Manufacturer, model, main features).
  12. When describing the variables contained in each formula, please point out the units using the International Units System. They are exposed in Table 1, but this is already a practical information.
  13. Please include a Discussion section where you cross your results with others previously obtained.
  14. Please become your Conclusions soundness.

Best wishes.

Kind regards.

Author Response

(The authors gave the same response as above.)

Round 2

Reviewer 2 Report

Authors corrected the paper as per reviewer comment, so I can suggest its publishing without any further corrections.

Reviewer 3 Report

This work remains a purely experimental work as the theoretical part is almost nil. For example, the explanations given by the authors are not valid, since friction is dictated by the roughness of the mating surfaces, their hardness, the rolling parameters (speed and load), and the lubrication regime. 

In answer to question 1, some phrases are added by copy-paste from existing literature, as evidenced by the incorrect hyphenation of words. Please proofread them.

Lubrication conditions influence the coefficient of friction and it is not the other way around as explained by the authors.

As for the calculation of the lubricant film thickness, I see that the authors have no idea and have assumed full HD lubrication, and this assumption is false.

However, I appreciate the authors' efforts and consider the experimental results valuable. 

This is the reason why I recommend the publication of this paper.

Reviewer 4 Report

Good improvement. Congratulations.

This manuscript is a resubmission of an earlier submission. The following is a list of the peer review reports and author responses from that submission.

Round 1

Reviewer 1 Report

This paper discussed measurement results of spindle noise due to variation of tip surface coating. Measurement data can be a valuable reference to reduce noise influence in various industries, e.g., textile and garment. Numbers of manuscript parts need attention for draft revision, i.e.:

  1. The first paragraph of Section 1 must be supported by data and reliable references.
  2. The data of the experimental instrument is not complete. Make sure to put manufacturer name, city, country for instruments, e.g., sensors, etc.; and purity percentage, manufacture name, city, country for a compound.
  3. The fringe or color level from blue to red in Figure 3 is not clear. No numbers are attached so the significance of the colors in this figure cannot be concluded. See Figure 6 for example.
  4. The experimental setup in Figure 4 is not clear and hard to be replicated for sake of benchmarking. The technical drawing needs to be added, including distance and exact location of sensor position and involved components in this work.
  5. Symbols in explanation for Equation has not used italic style. Furthermore "0" in the explanation has to be lower scripted.
  6. What is the purpose to use the bold style in the equation? Please refer to the journal template.
  7. What is the meaning of 10-4 s in Page 4? Do you mean 10^-4 s? Check typos in the manuscript.
  8. What is the deployed software to generate results as presented in Figures 5-9?
  9. Please also summarize all experimental data in form of a table so that which one is superior and which one is inferior, including % significance of the data can be comfortably cross-checked.
  10. Add your recommendation for future work based on your findings. Especially for real-world applications in industries.
  11. Point out the best variation in conclusion.
  12. It is indeed the friction coefficient has no unit. Therefore, authors need to write (-) after the name of the dimension.
  13. Add references from Coating.

Reviewer 2 Report

This submission cannot be accepted for publication for several reasons:

  1. The noise is reduced by not more than 1.5 %, this does not warrant a high experimental level.
  2. The chemical composition of the coatings is not introduced. What is SP2\SP3\C+ ?
  3. What are YD4203 spindles?
  4. The English is such that some sentences cannot be understood

Reviewer 3 Report

This paper presents the results of measurements of the sound pressure level (SPL) assumed to be produced by the contact between the rotor tip and the stator during operation of the textile equipment over a wide speed range (1 000-21 000 rpm). The tips of three sample rotors were coated by surface plating technology with Al-Ti-Cr-C, Ti-C, and SP2\SP3\C+. In terms of SPL, measurements show better results for Ti-C coating, but no scientific basis for the work is provided. I am tempted to reject this paper, but I will offer the authors some suggestions that might bring this paper up to the required publication standards:

  1. The equipment is not shown because the reference [24] cannot be found on the internet. Almost all previous works of the authors are published in Chinese journals and are not accessible to readers. Please provide full information on the test platform, equipment, data acquisition board, sensor, manufacturers, parameters, etc.
  2. The manufacturer of the powder is not mentioned. Also, the main properties of the powders and the proportions of the components are not specified.
  3. The deposition parameters of the surface cladding technology are not revealed. 
  4. Roughness values of the uncoated and coated rotor tips are not specified.
  5. Did the authors measure the friction coefficient values reported in Table 1? The coefficient of friction depends on so many factors that the recommended values without any experiment are not acceptable.
  6. Please describe the hardness and coating thickness measuring equipment.
  7. For the results shown in Figure 5b, the FFT application does not keep the signal in the time domain but in the frequency domain and the axis scale does not remain the same. In my opinion, smoothing filtering rather than FFT was applied.
  8. Please explain the phrase in lines 151-154 as it seems nonsense.
  9. There are many speculative sentences on lubrication conditions, e.g. lines 156 and 204, and friction coefficient values e.g. lines 175-178. Please present the results of the friction tests and calculate the lubrication parameter based on the calculation of the film thickness and the actual values of the roughness of the mating surfaces, also specifying the viscosity of the lubricant (for hydrodynamic lubrication this is a mandatory condition). 
  10. To prove that metal carbide and diamond-like carbon film are formed, EDS analysis and SEM images of the worn surfaces must be presented (see lines 202-205).
  11. The last paragraph before the Conclusions section contradicts the authors' previous results. In [7] was found that "The lower the surface roughness, the lower the friction noises". The new finding has no scientific basis.
  12. The "cherry on top" of this research is the conclusion that the "coating Ti-C containing only the two main elements of Ti and C produces the smallest sound pressure level in the experiment speed range". The reduced number of chemical elements produced less noise!

I realize that the whole paper needs to be redrafted and much more research is needed. For this reason, I decided to reject this paper, but I advise the authors to revise it as suggested and to resend it to Coatings journal.